# Distributed Dynamic Event-Triggered Control to Leader-Following Consensus of Nonlinear Multi-Agent Systems with Directed Graphs

**DOI:** 10.3390/e26020113

**Published:** 2024-01-26

**Authors:** Jia-Cheng Guan, Hong-Wei Ren, Guo-Liang Tan

**Affiliations:** 1College of Automation, Guangdong University of Petrochemical Technology, Maoming 525000, China; gjc18362938963@163.com (J.-C.G.); tan712999@163.com (G.-L.T.); 2College of Information and Control Engineering, Jilin Institute of Chemical Technology, Jilin 132022, China; 3School of Automation, Guangdong University of Technology, Guangzhou 510006, China

**Keywords:** nonlinear multi-agents, dynamic event-triggered, leader-following tracking, directed communication networks, Zeno behavior

## Abstract

This paper investigates achieving leader-following consensus in a class of multi-agent systems with nonlinear dynamics. Initially, it introduces a dynamic event-triggered strategy designed to effectively alleviate the strain on the system’s communication resources. Subsequently, a distributed control strategy is proposed and implemented in the nonlinear leader-follower system using the dynamic event-triggered mechanism, aiming to ensure synchronization across all nodes at an exponential convergence speed. Thirdly, the research shows that under the dynamic event-triggered strategy the minimum event interval of any two consecutive triggers guarantees the elimination of Zeno behavior. Lastly, the validity of the calculation results is verified by a simulation example.

## 1. Introduction


The consensus problem, recognized as one of the central problems in cooperative control, has garnered significant attention in recent decades, resulting in numerous noteworthy contributions. It has found applications in diverse domains such as wind power systems, distributed sensor networks, and blockchain technology, among others (refer to [1,2,3]). Ref. [4] is a representative pioneering work. In real-world application, all nodes in a Multi-Agent System (MAS) share the same communication network, which inevitably limits communication bandwidth and resources. Instead of relying on continuous signals, sampled signals provide an alternative approach. This allows each agent to utilize sampled signals, rather than continuous signals from its neighbors, for the control design [5,6]. Relying on continuous signals implies a need for communication networks with unlimited bandwidth and resources, which is clearly not practical in real-world applications. Therefore, various sampling control methods have been presented [7,8]. Notably, event-triggered control [9] demonstrates advantages in terms of low consumption of communication resources and practical application effectiveness. By employing this control method, signal sampling can be effectively reduced. In the case of periodic sampling, researchers can adaptively adjust the sampling frequency according to the system’s requirements.

Nowadays, communication networks are widely utilized, but they are sometimes confronted with the risk of various malicious attacks, such as DoS attacks [10] and FDI attacks [11]. Under the influence of network attacks, achieving continuous communication between agents becomes challenging. The event-triggered strategy is designed with proper triggering functions and control laws, enabling effective mitigation of the risks posed by cyber attacks [12]. Consequently, scholars are increasingly focusing on the study of event-triggered control mechanisms.

This control strategy has been widely applied to address consensus control in complex network systems and MASs with dynamics of different orders. Examples include first- and second-order dynamic models in [13,14,15,16], and higher-order models in [17,18,19]. There is no doubt that the choice of event-triggered strategy plays a decisive role in the performance of the system. What should be emphasized here is that traditional research on Event-Triggered Control (ETC) has predominantly centered around static event-triggering mechanisms. This implies that the measurement error is only affected by the state of the neighboring nodes. In such a static framework, the triggering condition is predefined and does not adapt to the system’s changing dynamics or performance metrics.

Recognizing these limitations, researchers have innovated various event-triggered control strategies, each characterized by unique features. Periodic Event-Triggered Control (PETC) [20], for instance, melds the attributes of both event-triggered and time-triggered controls. It periodically evaluates the system state to determine the need for control actions, thus achieving a balance between continuous monitoring and efficiency in response.

Self-Triggered Control [21] represents a proactive strategy where the controller not only executes control actions but also forecasts the timing for the next intervention. This approach significantly reduces the demand for real-time monitoring, thereby decreasing the overall system load.

Threshold-Triggered Control [22], on the other hand, operates based on specific threshold values. It initiates control signals once the system state surpasses these predetermined thresholds, enabling timely responses to critical state changes.

Model-Based Event-Triggered Control (MB-ETC) [23] leverages predictive modeling of system behavior. It triggers control actions when model predictions signal the need for intervention, offering a forward-looking control approach.

Adding to these developments, Dynamic Event-Triggered Control [24] has emerged as a significant innovation. This approach adapts the triggering conditions in real-time based on the system’s current state and performance, thereby offering greater flexibility and efficiency compared to static mechanisms.

Stochastic Event-Triggered Control [25] is especially effective for systems with inherent uncertainties and noise. It incorporates stochastic elements into the decision-making process, thus enhancing the robustness in unpredictable environments.

Furthermore, recent advances in machine learning and artificial intelligence have opened new avenues in ETC research. Researchers are exploring how these technologies can be employed to optimize event-triggering mechanisms, including the use of reinforcement learning [26] to determine the most effective triggering strategies.

Yin et al. presents a comparative study of distributed consensus gossip algorithms for network size estimation. The results are helpful in selecting the optimal consensus gossip algorithm for network size estimation [27]. Yan et al. introduces the adaptive memory-event-triggered control into T-S fuzzy systems [28]. This method is more robust and effective in dealing with system uncertainty and interference. In [29], Girard et al. investigates a dynamic event-triggered strategy, adding dynamic variables to the trigger function to optimize the measurement error threshold. Moreover, the results show that this strategy has a relatively longer minimum event-triggered interval compared with the general static event-triggered policy. Hu et al. addressed the synchronization problem in linear systems with directed graphs [30]. Researchers have introduced the dynamic triggered strategy into complex dynamic networks with discrete time delay to solve the synchronization control problem [31]. Combining the concepts of a centralized event-triggered control and a distributed event-triggered control strategy, a new adjustable event-triggered control protocol was suggested by Chen et al., in [32]. Under the constraints of different actual scenarios, the optimal value of parameter ξ∈[0,1] can be selected in the adjustable event-triggered strategy to achieve lower communication consumption cost and more satisfactory system performance. In [33], a dynamic event-triggered strategy was introduced into the leader-following consensus control of linear systems.

Taking inspiration from these aforesaid papers, we investigated the leader-following consensus problem via a distributed dynamic event-triggered control protocol with nonlinear dynamics under directed graphs. The main contributions of this study are given as follows:

(1) Compared with the static event-triggered algorithms for most linear systems, our dynamic event-triggered consensus algorithm is designed for distributed dynamic event-triggered quality under directed graphs, which is more general.

(2) By this new control strategy, the consensus of leader-following in nonlinear dynamics is addressed. The results indicate that, through this control strategy, the nonlinear system eventually reaches the leader-following consensus with an exponential convergence rate.

(3) Through the contradiction method, each node in this nonlinear system has been demonstrated to not exhibit Zeno behavior under our dynamic event-triggered control mechanism. For nonlinear systems, proving Zeno-free behavior is considered to be very difficult.

The structure of this paper is as follows. In Section 2, several mathematical preliminaries and notations are presented. Section 3 introduces a distributed dynamic event-triggered control protocol with nonlinear dynamics under directed graphs. The section explores the exponential consensus and the Zeno-free behavior facilitated by the proposed event-triggered control protocol. Section 4 provides a numerical simulation to illustrate the advantages of the proposed scheme. Finally, Section 5 concludes the paper.

## 2. Preliminaries


### 2.1. Graph Theory and Some Supporting Results

Let *R*, ∥·∥ denote the set of all real numbers and the Euclidean norm, respectively. $R^{n\times n}$ and $R^{n}$ represent the set of n×n real matrices and *n*-dimensional Euclidean space. $1_{N}$ denotes a N×1 vector where all elements are set to 1. For a real symmetric matrix, *M*, we usually define $\lambda_{\min}\left(M\right)=\min\lambda_{i}\left(M\right)$ and $\lambda_{\max}\left(M\right)=\max\lambda_{i}\left(M\right)$, where $\lambda_{i}\left(M\right)$ represents any eigenvalue of *M*.

Suppose that $\tilde{A}\in R^{m\times n}$ and $\tilde{B}\in R^{p\times q}$. Then, we can form $\tilde{A}⊗\tilde{B}\in R^{mp\times nq}$, where ⊗ denotes the Kronecker product. In this paper, the following properties will be used:(1)$$\left(\tilde{A}⊗\tilde{B}\right)\left(\tilde{C}⊗\tilde{D}\right)=\left(\tilde{A}\tilde{C}⊗\tilde{B}\tilde{D}\right)\tilde{A}⊗\left(\tilde{B}+\tilde{C}\right)=\tilde{A}⊗\tilde{B}+\tilde{A}⊗\tilde{C}.$$

The first equation holds if and only if the number of columns in $\tilde{A}$ equals the number of rows in $\tilde{B}$ and the number of columns in $\tilde{C}$ equals the number of rows in $\tilde{D}$. The second equation holds if and only if $\tilde{B}$ and $\tilde{C}$ have equal numbers of rows and columns.

Consider the following graph, $\tilde{G}=\left(\hat{V},\hat{E},\hat{\Pi }\right)$, to represent the communication network. Here, $\hat{V}=\left\{0,\ldots ,N\right\}$ denotes the nodes, which form a finite nonempty set. $\hat{E}\subseteq \hat{V}\;\times \;\hat{V}$ represents the edge set and $\hat{\Pi }$ denotes the adjacency matrix, which is non-negative. Typically, the leader is represented by node 0, while the other nodes, V={1,…,N}, represent the followers.

Let G=(V,E,Π) represent the followers’ communication network, derived from $\tilde{G}$ by excluding the leader. E⊆V×V denotes the edge set, illustrating the communication links between nodes. If node *j* can transmit information to node *i*, represented by (j,i)∈E, then *j* is the in-neighbor of *i*; the set of all the in-neighbors of node *i* in graph G is defined as $N_{i}=\left\{j|\left(j,i\right)\in E\right\}$. For $\Pi =\left(\varpi ij\right)\in R^{N\times N}$, if (j,i)∈E, then $\varpi_{ij}=1$, and $\varpi_{ij}=0$ otherwise. Consider the Laplacian matrix, $L=\left(\ell_{ij}\right)\in R^{N\times N}$, of graph G. For each pair of nodes (i,j)∈E where i≠j, we set $\ell_{ij}=-1$; otherwise, $\ell_{ij}=0$. Additionally, $\ell_{ii}=-\sum \ell_{ij}$. Graph is undirected if (i,j)=(j,i); otherwise, it is directed.

Let $D=\mathrm{diag}\left\{d_{1},\cdots ,d_{N}\right\}$ represent the communication links between the leader node and node *i*. If the followers can receive signals from the leader node, set $d_{i}=1$; otherwise, $d_{i}=0$. Define H=L+D. A directed path in graph $\tilde{G}$ from node *i* to node *j* is defined as $\left(i,i_{1}\right),\left(i_{1},i_{2}\right),\ldots ,\left(i_{k},j\right)$. Graph $\tilde{G}$ is considered connected if there is a path from every node *i* in G to the leader node.

### 2.2. Problem Formulation

Consider a tracking problem under nonlinear dynamics, consisting of a leader node with *N* follower nodes. The dynamics of the *i*th follower nodes are defined as follows,
(2)$${\dot{x}}_{i}=Ax_{i}+Bu_{i}+Cf\left(x_{i}\right),i=1,\ldots ,N,$$
where $A\in R^{Nn\times Nn},B\in R^{Nn\times Nm}$, and $C\in R^{Nn\times Np}$; $x_{i}$, $u_{i}$ are the state vectors and control inputs associated with the *i*th nodes, respectively. The term $f\left(x_{i}\right)$ denotes the nonlinearity of the *i*th node. The dynamic of the isolated leader node is described by
(3)$${\dot{x}}_{0}\left(t\right)=Ax_{0}\left(t\right)+Cf\left(x_{0}\right).$$

To develop our proposed strategy, we introduce the following necessary assumptions.

**Assumption** **1.**
*Graph G is directed and strongly connected. Moreover, at least one follower node communicates with the leader.*


**Assumption** **2.**
*For all $\alpha_{i},\beta_{i}\in R^{n}$ where $\alpha_{i}\neq \beta_{i}$, there exist constants $L_{i},U_{i}$, representing the upper and lower bounds, respectively, which satisfies $L_{i}\leq \frac{f_{i}\left(\alpha_{i}\right)-f_{i}\left(\beta_{i}\right)}{\alpha_{i}-\beta_{i}}\leq U_{i},i=1,2,\ldots ,N$. Define $L_{g}=\max_{1\leq i\leq n}\left\{\left|L_{i}\right|,\left|U_{i}\right|\right\}$. It is straightforward to deduce that*



(4)
$$∥f_{i}\left(\alpha_{i}\right)-f_{i}\left(\beta_{i}\right)∥\leq L_{g}∥\alpha_{i}-\beta_{i}∥.$$


**Assumption** **3.**
*The pair (A,B) is stabilizable.*


Node *i* receives sampling information only at certain instants, $t_{0}^{i}$, $t_{1}^{i}$,…, i.e., the triggering time. For the current triggering instance, $t_{k}^{i}$ represents the instant when node *i* communicates with neighboring nodes. The following dynamic triggering mechanism determines the next triggering instant $t_{k+1}^{i}$,
(5)$$\left\{\begin{array}{c} t_{k+1}^{i}=\inf\left\{t>t_{k}^{i}|g_{i}^{d}\left(\underset{j\in N_{i}}{\cup }x_{j}\left(t\right),e_{i}\left(t\right),\rho_{i}\left(t\right)\right)\geq 0\right\}\\ {\dot{\rho }}_{i}\left(t\right)=h_{i}\left(e_{i}\left(t\right),q_{i}\left(t\right),\rho_{i}\left(t\right)\right) \end{array},\right.$$
where $\rho_{i}\left(t\right)$ denotes the internal dynamic state; $q_{i}\left(t\right)$ and $e_{i}\left(t\right)$ are to be defined later in Equation (Equation 6).

As illustrated by the above algorithm, the triggering function $g_{i}^{d}\left(\cdot \right)$ comprises the system state and internal dynamic $\rho_{i}$. Excluding internal dynamic $\rho_{i}$ reduces this strategy to a standard static event-triggering mechanism. The function $h_{i}\left(\cdot \right)$ describes the relationship between internal dynamics and system state.

**Lemma** **1.**
*Under Assumption 1, there exists $\tilde{\xi }={\left[{\tilde{\xi }}_{1},{\tilde{\xi }}_{2},\ldots ,{\tilde{\xi }}_{N}\right]}^{T}\in R^{N}$ such that ${\tilde{\xi }}^{T}H=0$, where all the elements are non-negative. Further, define $\Xi =\mathrm{diag}\left({\tilde{\xi }}_{1},{\tilde{\xi }}_{2},\ldots ,{\tilde{\xi }}_{N}\right)$, ${\tilde{\xi }}_{i}>0$, $\sum_{i=1}^{N}{\tilde{\xi }}_{i}=1$. Then, the matrix $\hat{H}=\left[\left(\Xi H+H^{T}\Xi \right)\right]/2$ is symmetric for i=1,…,N.*


**Lemma** **2**(Lemma 6 [34]). *Under Assumption 1, for a strongly connected network G and corresponding matrix H, its general algebraic connectivity, α(H), satisfies $\alpha \left(H\right)=\min_{x^{T}\tilde{\xi }=0,x\neq 0}\frac{x^{T}\hat{H}x}{x^{T}\Xi x}>0$, particularly $\alpha \left(H\right)=\lambda_{2}\left(H\right)$, when graph G is undirected.*

**Lemma** **3.**
*Under Assumption 1, the following Algebraic Riccati Equation (ARE) has a unique solution, P>0:*

(6)
$$PA+A^{T}P-PBB^{T}P+I_{n}=0.$$



## 3. Main
Results

In this section, the main results of our proposed strategies are described and derived.

Firstly, define the combined measurement variable $q_{i}\left(t\right)$ and the measurement error $e_{i}\left(t\right)$ as follows:(7)$$q_{i}\left(t\right)=\sum_{i=1}^{N}\varpi_{ij}\left(x_{j}\left(t\right)-x_{i}\left(t\right)\right)+d_{i}\left(x_{0}\left(t\right)-x_{i}\left(t\right)\right),e_{i}\left(t\right)=q_{i}\left(t_{k}^{i}\right)-q_{i}\left(t\right).$$

For node *i*, we define a continuous state feedback control protocol with triggered signal $q_{i}\left(t_{k}^{i}\right)$ as follows:(8)$$u_{i}\left(t\right)=Kq_{i}\left(t_{k}^{i}\right),\tau \in \left[t_{k}^{i},t_{k+1}^{i}\right).$$

Define the tracking error $\zeta_{i}\left(t\right)=x_{i}\left(t\right)-x_{0}\left(t\right)$. Arrange $q\left(t\right)=\mathrm{col}\left(q_{1}\left(t\right),\ldots ,q_{N}\left(t\right)\right)$, and note that $u_{i}\left(t\right),\zeta \left(t\right)$ and e(t) are in the similar forms. Define $\psi \left(x\right)=\mathrm{col}\left(f\left(x_{1}\right),\ldots ,f\left(x_{N}\right)\right)$. Note that $q\left(t\right)=-\left(H⊗I_{n}\right)\zeta \left(t\right)$, and in view of Equation (Equation 7) it follows that
(9)$$\dot{q}\left(t\right)=\left(I_{N}⊗A+H⊗BK\right)q\left(t\right)+\left(H⊗BK\right)e\left(t\right)+\left(H⊗C\right)\Theta \left(x\right),$$
where $\Theta \left(x\right)=\mathrm{col}\left(f\left(x_{1}\right)-f\left(x_{0}\right),\ldots ,f\left(x_{N}\right)-f\left(x_{0}\right)\right)$.

The solution matrix $P=P^{T}>0$ of the following equation can be obtained from Lemma 3,
(10)$$PA+A^{T}P+{\tilde{\xi }}_{M}PCC^{T}P+L_{g}^{2}{{\tilde{\xi }}_{m}}^{-1}I_{N}-\Lambda^{\prime }PBB^{T}P\leq -I_{N},$$
where $\Lambda^{\prime }=\left(2\sigma_{i}a\left(H\right)-\sigma_{i}^{2}{\tilde{\xi }}_{i}∥H∥\right)>0$, with a(H) defined in Lemma 2, and the other parameters will be determined later.

The main result of our proposed strategy will be summarized in the following proof.

**Theorem** **1.**
*Under both Assumptions 1 and 2, $K=\sigma B^{T}P$, where $0<\sigma <\left(2a\left(H\right)/\left[{\tilde{\xi }}_{M}{∥H∥}^{2}\right]\right)$. Therefore, all the nodes in this MAS (1) achieve exponential consensus with $t_{k+1}^{i}$, determined by the following event-triggered strategy,*


(11)$$\left\{\begin{array}{c} t_{k+1}^{i}=\inf\left\{t>t_{k}^{i}|{∥e_{i}\left(t\right)∥}^{2}-\delta_{i}{∥q_{i}\left(t\right)∥}^{2}-\tilde{\sigma_{i}}\rho_{i}\left(t\right)\geq 0\right.\\ {\dot{\rho }}_{i}\left(t\right)=-\beta_{i}\rho_{i}\left(t\right)-\alpha_{i}\left({∥e_{i}\left(t\right)∥}^{2}-\delta_{i}{∥q_{i}\left(t\right)∥}^{2}\right), \end{array}\right.$$
where we let the parameters $\tilde{\sigma_{i}}\geq 0$, $\beta_{i}\geq 0$, $\alpha_{i}\geq ∥PBB^{T}P∥$, $\delta_{i}\leq \tilde{\xi }/\alpha_{i}$ under our proposed event-triggered block (11). Zeno behavior is eliminated for any nodes in nonlinear MASs.

**Proof.** Consider a Lyapunov function candidate as follows:
(12)$$V\left(t\right)=q^{T}\left(t\right)\left(\Xi ⊗P\right)q\left(t\right)+\sum_{j=1}^{N}\rho_{i}\left(t\right).$$According to event-triggered block (11), it satisfies ${∥e_{i}\left(t\right)∥}^{2}-\delta_{i}{∥q_{i}\left(t\right)∥}^{2}\leq \tilde{\sigma_{i}}\rho_{i}\left(t\right)$, which means
(13)$${\dot{\rho }}_{i}\geq -\beta_{i}\rho_{i}\left(t\right)-\alpha_{i}\tilde{\sigma_{i}}\rho_{i}\left(t\right).$$Using the comparison principle, we obtain
(14)$$\rho_{i}\left(t\right)\geq \rho_{i}\left(0\right)e^{-\left(\beta_{i}+\alpha_{i}\tilde{\sigma_{i}}\right)t}>0,$$
which guarantees that V(t)>0.Since $∥f_{i}\left(\alpha_{i}\right)-f_{i}\left(\beta_{i}\right)∥\leq L_{g}∥\alpha_{i}-\beta_{i}∥$, under the definition of nonlinearity in Assumption 2, it is straightforward to deduce that
(15)$$∥\sum_{j=1}^{N}c_{ij}\left(f\left(x_{j}\right)-f\left(x_{i}\right)\right)∥\leq L_{g}\sum_{j=1}^{N}c_{ij}\left(x_{j}-x_{i}\right),$$
where $L_{g}=\max_{1\leq i\leq n}\left\{\left|L_{i}\right|,\left|U_{i}\right|\right\}$. Then, we can readily derive the following properties:
(16)$$\sum_{i=1}^{N}{\left(\sum_{j=1}^{N}c_{\mathrm{ij}}\left(f(x_{j}\right)-f\left(x_{i}\right)+d_{i}\left(f\left(x_{0}\right)-f\left(x_{i}\right)\right)\right)}^{2}\leq L_{g}^{2}\sum_{i=1}^{N}{\left(\sum_{j=1}^{N}c_{\mathrm{ij}}\left(x_{j}-x_{i}\right)+d_{i}\left(x_{0}-x_{i}\right)\right)}^{2}.$$Finally, we obtain
(17)$$∥\sum_{j=1}^{N}c_{ij}\left(f\left(x_{j}\right)-f\left(x_{i}\right)\right)+d_{i}\left(f\left(x_{0}\right)-f\left(x_{i}\right)\right)∥\leq L_{g}∥\sum_{j=1}^{N}c_{ij}\left(x_{j}-x_{i}\right)+d_{i}\left(x_{0}-x_{i}\right)∥.$$Next, considering Lemma 2, and substituting ${\tilde{\xi }}_{M}=\max_{i=1,\ldots ,N}\left\{{\tilde{\xi }}_{i}\right\}$, Equation (Equation 8) is employed with $K=\sigma B^{T}P$, along with the following constraints:
(18)$$\begin{array}{c} 2q^{T}\left(t\right)\left(\Xi H⊗PC\right)\Theta \left(x\right)\leq {\tilde{\xi }}_{M}q^{T}\left(t\right)\left(\Xi ⊗PCC^{T}P\right)\\ \times q^{T}\left(t\right)+\Theta^{T}\left(x\right)\left(H^{T}H⊗I_{n}\right)\Theta \left(x\right),\\ 2q^{T}\left(t\right)\left(\Xi H⊗\sigma PBB^{T}P\right)e\left(t\right)\leq q^{T}\left(t\right)\left(\Xi^{2}⊗∥H∥\sigma^{2}\right.\\ \left.\times \hat{B}\right)q^{T}\left(t\right)+e^{T}\left(t\right)\left(I_{N}⊗\hat{B}\right)e^{T}\left(t\right). \end{array}$$Let $\hat{B}=PBB^{T}P$. We can then express the following inequality:
(19)$$\begin{array}{cc} \dot{V}\left(t\right)\leq & q^{T}\left(t\right)\left(\Xi ⊗\left(PA+A^{T}P-2\sigma a\left(H\right)\hat{B}\right)\right)q\left(t\right)\\ \; & +\Theta^{T}\left(x\right)\left(H^{T}H⊗I_{n}\right)\Theta \left(x\right)+q^{T}\left(t\right)\left(\Xi^{2}⊗∥H∥\sigma^{2}\hat{B}\right)q\left(t\right)\\ \; & +{\tilde{\xi }}_{M}q^{T}\left(t\right)\left(\Xi ⊗PCC^{T}P\right)q\left(t\right)+e^{T}\left(t\right)\left(I_{N}⊗\hat{B}\right)e\left(t\right)+\rho_{i}\left(t\right). \end{array}$$Based on the definition of Θ(x) and Equation (Equation 14), we can derive:
(20)$$\begin{array}{c} \left(H⊗I_{n}\right)\Theta \left(x\right)=\left[\begin{array}{c} \sum_{j=1}^{N}c_{1j}\left(f\left(x_{j}\right)-f\left(x_{1}\right)+d_{i}\left(f\left(x_{0}\right)-f\left(x_{1}\right)\right)\right.\\ \vdots \\ \sum_{j=1}^{N}c_{Nj}\left(f\left(x_{j}\right)-f\left(x_{N}\right)+d_{i}\left(f\left(x_{0}\right)-f\left(x_{N}\right)\right)\right. \end{array}\right], \end{array}$$$\Theta^{T}\left(x\right)\left(H^{T}H⊗I_{n}\right)\Theta \left(x\right)=\sum_{i=1}^{N}{∥\sum_{j=1}^{N}c_{\mathrm{ij}}\left(f(x_{j}\right)-f\left(x_{i}\right)+d_{i}\left(f\left(x_{0}\right)-f\left(x_{i}\right)\right)∥}^{2}$.Then, we have
(21)$$\begin{array}{cc} \Theta^{T}\left(x\right)\left(H^{T}H⊗I_{n}\right)\Theta \left(x\right) & =L_{g}^{2}{\tilde{\xi }}_{m}^{-1}\sum_{i=1}^{N}{\tilde{\xi }}_{i}{∥\sum_{j=1}^{N}c_{ij}\left(x_{j}-x_{i}+x_{0}-x_{i}\right)∥}^{2}\\ \; & \leq L_{g}^{2}{\tilde{\xi }}^{-1}\sum_{i=1}^{N}{\tilde{\xi }}_{i}q_{i}^{T}\left(t\right)q_{i}\left(t\right). \end{array}$$By considering ${\tilde{\xi }}_{m}=\min_{i=1,\ldots ,N}\left\{{\tilde{\xi }}_{i}\right\}$ and substituting Equation (Equation 17) into Equation (Equation 16), we obtain:
(22)$$\begin{array}{cc} \dot{V}\left(t\right) & \leq q\left(t\right)(\Xi ⊗\left(2A-2\sigma a\left(H\right)\hat{B}\right))q\left(t\right)\\ \; & +{\tilde{\xi }}_{M}q^{T}\left(t\right)\left(\Xi ⊗PCC^{T}P\right)q\left(t\right)\\ \; & +L_{g}^{2}{\tilde{\xi }}_{m}^{-1}\sum_{i=1}^{N}\xi_{i}q_{i}^{T}\left(t\right)q_{i}\left(t\right)\\ \; & +q^{T}\left(t\right)\left(\Xi^{2}⊗∥H∥\sigma^{2}\hat{B}\right)q\left(t\right)\\ \; & +e^{T}\left(t\right)\left(I_{N}⊗\hat{B}\right)e\left(t\right)+\sum_{i=1}^{N}{\dot{\rho }}_{i}\left(t\right) \end{array}.$$Additionally, we have the following inequality:
(23)$$\begin{array}{c} \dot{V}\left(t\right)\leq -\sum_{i=1}^{N}{\tilde{\xi }}_{i}\left(1-\delta_{i}\alpha_{i}\right){∥q_{i}\left(t\right)∥}^{2}-\sum_{i=1}^{N}\beta_{i}\rho_{i}\left(t\right)-\left(\alpha_{i}-∥\hat{B}∥\right){∥e_{i}\left(t\right)∥}^{2}. \end{array}$$Substituting $\alpha_{i}\geq ∥\hat{B}∥,{\tilde{\xi }}_{i}\geq \delta_{i}\alpha_{i},{\hat{\theta }}_{M}=\max_{i=1,\ldots ,N}\left\{{\hat{\theta }}_{i}\right\},\beta_{m}=\min_{i=1,\ldots ,N}\left\{\beta_{i}\right\}$ into Equation (Equation 20), we finally obtain,
(24)$$\begin{array}{cc} \dot{V}\left(t\right) & \leq -\sum_{I=1}^{N}{\tilde{\xi }}_{i}\left(1-{\hat{\theta }}_{M}\right){∥q_{i}\left(t\right)∥}^{2}-\beta_{m}\sum_{i=1}^{N}\rho_{i}\left(t\right)\\ \; & \leq -\left(\left(1-{\hat{\theta }}_{M}\right)/\lambda_{\max}\left(P\right)\right)q^{T}\left(t\right)\left(\Xi ⊗P\right)q\left(t\right)-\beta_{m}\sum_{i=1}^{N}\rho_{i}\\ \; & \leq -\varepsilon_{0}V\left(t\right), \end{array}$$
where $\varepsilon_{0}=\min\left[\left(1-{\hat{\theta }}_{M}\right)/\lambda_{\max}\left(P\right),\beta_{m}\right]>0$. The consensus ultimately reaches exponential synchronization at decay rate $\varepsilon_{0}$. □

**Remark** **1.**
*Our strategy can guarantee leader-following synchronization, as demonstrated in the proof of Theorem 1, which is not covered in [30,33,35,36]. Clearly, our strategy is more general.*


**Remark** **2.**
*The strategy proposed in Theorem 1 addresses the issue of dynamic event-triggering under nonlinear conditions. In most existing event-triggered strategies, such as those in [34,37], the interference of nonlinear dynamics is seldom considered. Due to the limited availability of effective tools for handling nonlinearity, research on nonlinear event-triggered consensus remains scarce and presents a challenging and open question.*


**Remark** **3.**
*In contrast to the similar consensus strategies proposed in [38,39], our strategy focuses on dynamic event-triggered consensus rather than static consensus. Furthermore, the proposed strategy guarantees exponential convergence and can be applied in scenarios involving directed graphs.*


**Remark** **4.**
*The proposed dynamic ETM incorporates an adaptive thresholding mechanism, allowing the triggering threshold to dynamically adjust based on the system’s current state. This adaptive feature enables optimized event-triggering decisions, reducing unnecessary triggering events while ensuring system stability and performance. We conducted a performance evaluation of this proposed dynamic ETM, comparing it to existing mechanisms. Through extensive simulations and experimental analysis, we have demonstrated the effectiveness and superiority of this proposed mechanism in terms of improved system performance, reduced triggering events, and enhanced resource utilization.*


The following is used to verify that Zeno behavior will not occur for any nodes under our strategy.

**Definition** **1.**
*Zeno behavior in MASs is ruled out if a finite number of communication events are satisfied between MASs in any finite time period. Specifically, for every node i, the sequence of triggers, $t_{k}^{i}$, satisfies the property $\underset{k>0}{\inf}\left(t_{k+1}^{i}-t_{k}^{i}\right)>0.$*


**Proof.** Assuming that the node *i* incurs Zeno behavior at $\tilde{t}$. Given that $\lim_{k\to \infty }t_{k}^{i}=\tilde{t}$, for any ϵ>0 there exists N(ϵ) such that $t_{k}^{i}\in \left(\tilde{t}-\epsilon ,\tilde{t}+\epsilon \right)$ for $\forall k\geq N\left(\epsilon \right)$, indicating that $t_{N(\epsilon )+1}^{i}-t_{N(\epsilon )}^{i}<2\epsilon$. The norm $∥e_{i}\left(t\right)∥$ is piecewise continuous and differentiable in $\left[t_{k}^{i},t_{k+1}^{i}\right)$. Therefore, the Dini derivative of $∥e_{i}\left(t\right)∥$ yields
(25)$$\begin{array}{cc} D^{+}∥e_{i}\left(t\right)∥ & \leq \frac{∥e_{i}^{T}∥}{∥e_{i}∥}∥{\dot{e}}_{i}∥=∥-\sum_{j=1}^{N}\varpi_{ij}\left({\dot{x}}_{j}\left(t\right)-{\dot{x}}_{i}\left(t\right)+d_{i}\left({\dot{x}}_{0}\left(t\right)-{\dot{x}}_{i}\left(t\right)\right)∥\right.\\ \; & =∥-Aq_{i}\left(t\right)-\sigma BB^{T}P\sum_{j=1}^{N}\varpi_{ij}\left(q_{i}\left(t_{k}^{i}\right)-q_{i}\left(t_{k}^{j}\right)\right)+d_{i}\left(q_{0}\left(t_{k}^{o}\right)-q_{i}\left(t_{k}^{i}\right)\right)∥\\ \; & +∥C\sum_{j=1}^{N}\varpi_{ij}\left(f\left(x_{i}\right)-f\left(x_{j}\right)\right)+d_{i}\left(f\left(x_{0}\right)-f\left(x_{i}\right)\right)∥ \end{array}.$$From stability analysis, we have $∥q_{i}\left(t\right)∥\leq \sqrt{\frac{V(0)}{{\tilde{\xi }}_{m}\lambda_{\min}\left(P\right)}}$. Then, the upper bound can be calculated as follows:
(26)$$\begin{array}{c} D^{+}∥e_{i}\left(t\right)∥\leq \left(∥A∥+L_{g}∥C∥+\sigma ∥BB^{T}P∥\left(1+\left|N_{i}\right|\right))∥q_{i}\left(t\right)∥\right.,\\ \sqrt{\frac{V(0)}{{\tilde{\xi }}_{m}\lambda_{\min}\left(P\right)}}≐{\hat{V}}_{0}. \end{array}$$From Equation (Equation 11), it is derived that $∥e_{i}\left(t\right)∥\geq \sqrt{\delta_{i}∥q_{i}{\left(t\right)}^{2}∥+\tilde{\sigma_{i}}\rho_{i}}=\sqrt{\tilde{\sigma_{i}}\rho_{i}}$. Therefore, $∥e_{i}\left(t_{k}^{i-}\right)∥\geq \sqrt{\tilde{\sigma_{i}}\rho_{i}\left(t_{k}^{i-}\right)}=\sqrt{\tilde{\sigma_{i}}\rho_{i}\left(0\right)}e^{-\frac{\beta_{i}+\alpha_{i}\tilde{\sigma_{i}}}{2}t_{k}^{-}}$, which leads to $t_{N\left(\epsilon \right)+1}^{i}-t_{N\left(\epsilon \right)}^{i}\geq$$\frac{1}{{\hat{V}}_{0}}\sqrt{\tilde{\sigma_{i}}\rho_{i}\left(0\right)}e^{-\frac{\beta_{i}+\alpha_{i}\tilde{\sigma_{i}}}{2}t_{N\left(\epsilon \right)+1}^{i}}$. When ϵ>0, it can be shown that
(27)$$\frac{1}{{\hat{V}}_{0}}\sqrt{\tilde{\sigma_{i}}\rho_{i}\left(0\right)}e^{-\frac{\beta_{i}+\alpha_{i}\tilde{\sigma_{i}}}{2}T_{0}}=2\epsilon e^{\frac{\beta_{i}+\alpha_{i}\tilde{\sigma_{i}}}{2}\epsilon }.$$The conditions in Equation (Equation 25) clearly contradicts the fact that
(28)$$t_{N\left(\epsilon \right)+1}^{i}-t_{N\left(\epsilon \right)}^{i}\geq \frac{1}{{\hat{V}}_{0}}\sqrt{\tilde{\sigma_{i}}\rho_{i}\left(0\right)}e^{-\frac{\beta_{i}+\alpha_{i}\tilde{\sigma_{i}}}{2}\left(T_{0}+\epsilon \right)}=2\varepsilon ,$$
which indicates that the Zeno behavior will not occur for node *i*. The proof is completed. □

**Remark** **5.**
*Zeno behavior is an abnormal phenomenon that often arises in event-triggered systems or hybrid systems, signifying infinite cumulative execution within a finite amount of time. It may occur due to the design of event-triggered conditions or specific system dynamics. Investigating Zeno behavior is crucial for gaining a comprehensive understanding of potential issues related to system stability, making it a challenging research topic.*


**Remark** **6.**
*Compared to the triggering strategy in [38], our approach takes into account the dynamic variable $\rho_{i}\left(t\right)$. If $\rho_{i}\left(t\right)\equiv 0$, the dynamic event-triggered control proposed in this paper reverts to the static event-triggered control in [38]. When $\tilde{\sigma_{i}}\rho_{i}\left(t\right)>0$, simulation results demonstrate that the dynamic triggering strategy proposed in this paper ensures a more efficient time interval compared to the static strategy presented in [38]. This improvement significantly reduces the likelihood of Zeno behavior.*


**Remark** **7.**
*Designing a feasible and effective dynamic event-triggered strategy while ensuring the absence of Zeno behavior is undoubtedly a formidable challenge. As is evident from the findings in [9,40,41], the development of practical event-triggered strategies is intricately tied to the solutions of certain matrix inequalities. Ensuring their existence is no small feat. It is worth noting that feasible matrix gains can be derived from the unique solutions of the defined Algebraic Riccati Equation (ARE). Furthermore, as demonstrated in the proof above, a reductio ad absurdum argument effectively addresses this challenge.*


## 4. Illustrative Example


To validate the feasibility of the aforementioned theory, in this section we present a simulation.

**Example** **1.**
*Consider a nonlinear system comprising eight agents, where the model of spacecraft formation flying dynamics describes the dynamic of all nodes. The linearized equation for the ith spacecraft is given as follows:*



(29)
$$\begin{array}{cc} {\ddot{\bar{x}}}_{i}-2\pi_{0}{\dot{\bar{y}}}_{i} & =u_{xi}\\ {\ddot{\bar{y}}}_{i}+2\pi_{0}{\dot{\bar{x}}}_{i}-3\pi_{0}^{2}{\bar{y}}_{i} & =u_{yi}\\ {\ddot{\bar{z}}}_{i}+\pi_{0}^{2}{\dot{\bar{z}}}_{i} & =u_{zi} \end{array}.$$


Each node’s state is represented by $x_{i}=\mathrm{col}\left({\bar{x}}_{i},{\bar{y}}_{i},{\bar{z}}_{i},v_{i}^{x},v_{i}^{y},v_{i}^{z}\right)$. Here, $\left({\bar{x}}_{i},{\bar{y}}_{i},{\bar{z}}_{i}\right)$ denotes the position of the *i*-th node in the X–Y–Z directions, while $\left(v_{i}^{x},v_{i}^{y},v_{i}^{z}\right)$ and $\left(u_{xi},u_{yi},u_{zi}\right)$ represent the velocity and the control input for the *i*-th agent, respectively. Transforming the spacecraft formation model into a nonlinear consensus problem is a straightforward task. Consensus is deemed to have been achieved when the velocity states converge to the same values and the position states converge towards anticipated synchronization. In other words, as t→∞, we have $h_{i}-p_{i}\to h_{j}-p_{j}$ and $\dot{h}i\to \dot{h}j$. Here, $p_{i}-p_{j}\in R^{3}$ represents the anticipated separation between agents *i* and *j*.
$$\begin{array}{cc} A & =\left[\begin{array}{cc} 0 & I_{3}\\ \hat{A} & \tilde{A} \end{array}\right],B=\left[\begin{array}{c} 0\\ I_{3} \end{array}\right],C=I\\ \hat{A} & =\left[\begin{array}{ccc} 0 & 0 & 0\\ 0 & 3\pi_{0}^{2} & 0\\ 0 & 0 & -\pi_{0}^{2} \end{array}\right],\tilde{A}=\left[\begin{array}{ccc} 0 & 2\pi_{0} & 0\\ -2\pi_{0} & 0 & 0\\ 0 & 0 & 0 \end{array}\right],f\left(x_{i}\right)={\left[\begin{array}{cccccc} 0 & 0 & 0 & 0 & 0 & 0.2\sin x_{i4} \end{array}\right]}^{T} \end{array}$$

Here, $\pi_{0}=0.001$ represents the agent’s angular rate. The communication graph is provided in Figure 1. It can be deduced from Lemma 1 that

$\tilde{\xi }={\left[\begin{array}{cccccccc} 0.1886 & 0.0773 & 0.0386 & 0.1955 & 0.1205 & 0.1659 & 0.1341 & 0.0795 \end{array}\right]}^{T}$, which satisfies the condition ${\tilde{\xi }}^{T}H=0$ and $\sum_{i=1}^{N}{\tilde{\xi }}_{i}=1$. The other parameters are chosen accordingly.



$$\begin{array}{c} a\left(H\right)=0.9661,\;\sigma =0.0785\leq \frac{2a(H)}{{\tilde{\xi }}_{M}{∥H∥}^{2}}=0.3522,\\ B^{T}P=\left[\begin{array}{cccccc} 3.2163 & -0.0081 & 0 & 8.7721 & 0 & 0\\ 0.0081 & 3.2163 & 0 & 0 & 8.7721 & 0\\ 0 & 0 & 3.2163 & 0 & 0 & 8.7721 \end{array}\right],\\ \delta =\left[\begin{array}{cccccccc} 1.4 & 0.6 & 0.6 & 1.2 & 1.0 & 0.8 & 1.2 & 0.8 \end{array}\right]\times 10^{-3},\\ \alpha_{i}=∥\hat{B}∥=87.3971,\;\beta_{i}=0.004,\;\tilde{\sigma_{i}}=0.002. \end{array}$$



$e_{ij}={\bar{x}}_{i}-{\bar{x}}_{j},\forall i\neq j$ denotes the position states. Figure 2 depicts the trajectory of the position errors ($e_{ij}$), Figure 3 illustrates the velocity trends of the nodes, and Figure 4 shows the trajectory of the control inputs for the nodes. Additionally, Figure 5 represents the trajectory of the position errors of the nodes. As shown in the figure, both the position and the velocity states of all nodes eventually converge to an anticipatory synchronization.

Figure 2 displays a set of spacecraft position states, illustrating the trajectory of position states influenced by a nonlinear environment (In this article, a cosine function is employed to model a potential nonlinear environment). It can be observed that as time progresses the system’s position state trajectory becomes more consistent and exhibits stability with periodic motion.

Figure 6 and Table 1 clearly illustrate the inter-event duration and the numbers of triggering events. Figure 7 displays the triggering thresholds and the combined measurement errors for Cases 1 and 2. The results demonstrate that all nodes achieve synchronization, thereby achieving consensus.

For comparison, a simulation was also conducted using a static event-triggered approach [42]. The trigger sequences for nodes using the dynamic trigger mechanism (case 1) and the static trigger mechanism (case 2) are recorded respectively in Table 1. Additionally, Figure 7 compares the triggering threshold and the error evolution curves. It can be observed that the dynamic triggering mechanism significantly reduces the number of triggers.

## 5. Conclusions

In this paper, our focus was on addressing the leader-following consensus problem using a distributed dynamic event-triggered control protocol with nonlinear dynamics under directed graphs. To some extent, this paper can be seen as an extension of References [27,30]. Our results demonstrate that the proposed control strategy can achieve exponential synchronization and, importantly, ensures that Zeno behavior does not occur for any node. However, there remain significant open challenges in this field, and they require further investigation. To date, there has been limited research effort dedicated to studying cyber attacks within the context of dynamic event-triggered control. In our future research endeavors, we will explore dynamic event-triggered control strategies for mitigating cyber attacks and addressing time-varying dynamics.

## Figures and Tables

**Figure 1 entropy-26-00113-f001:**
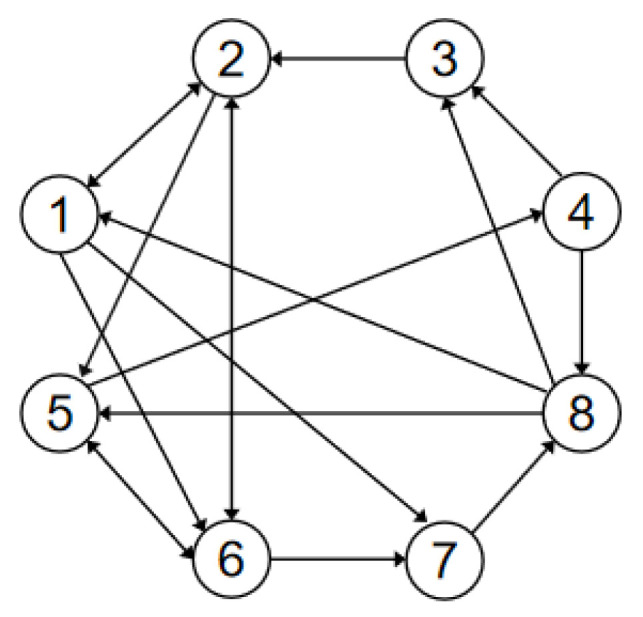
Communication graph topology of nodes.

**Figure 2 entropy-26-00113-f002:**
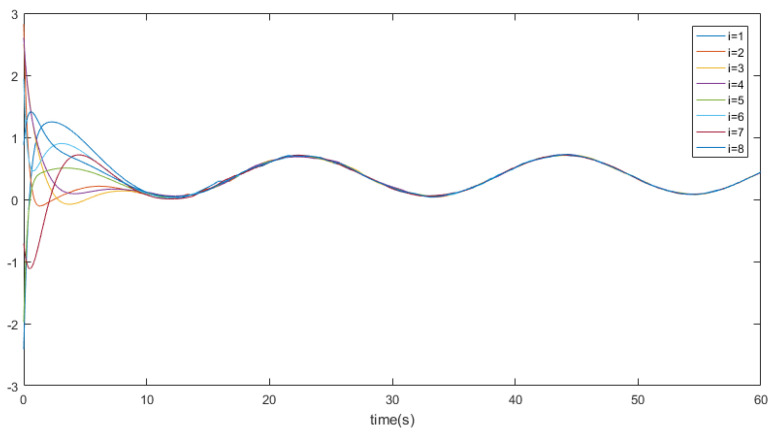
The trajectory curves of nodes’ position states.

**Figure 3 entropy-26-00113-f003:**
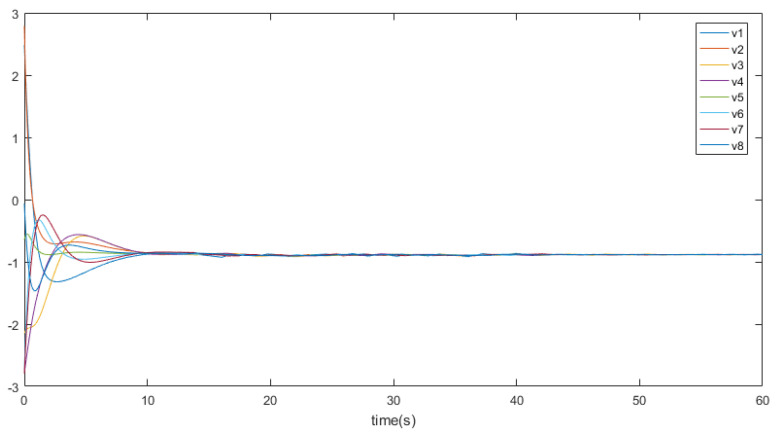
The trajectory curves of nodes’ velocity states.

**Figure 4 entropy-26-00113-f004:**
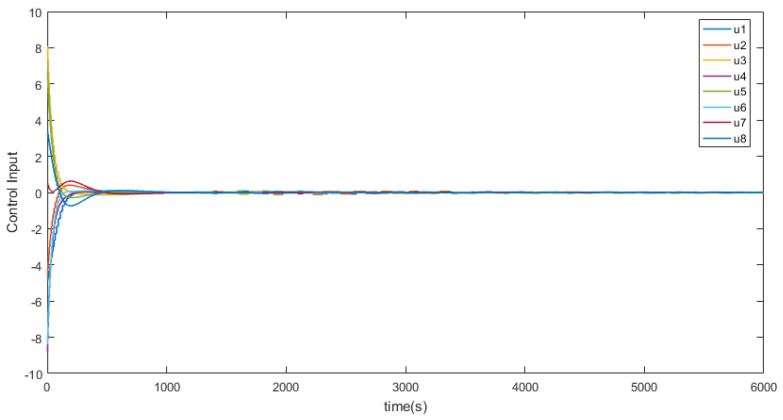
The trajectory curves of nodes’ control inputs.

**Figure 5 entropy-26-00113-f005:**
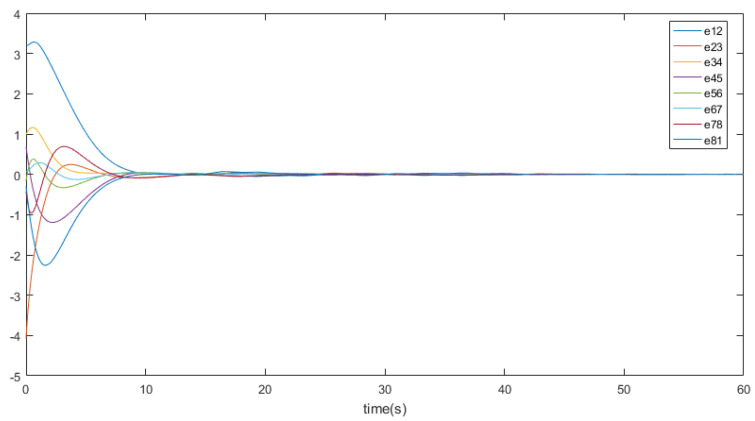
The trajectory curves of the position errors of nodes.

**Figure 6 entropy-26-00113-f006:**
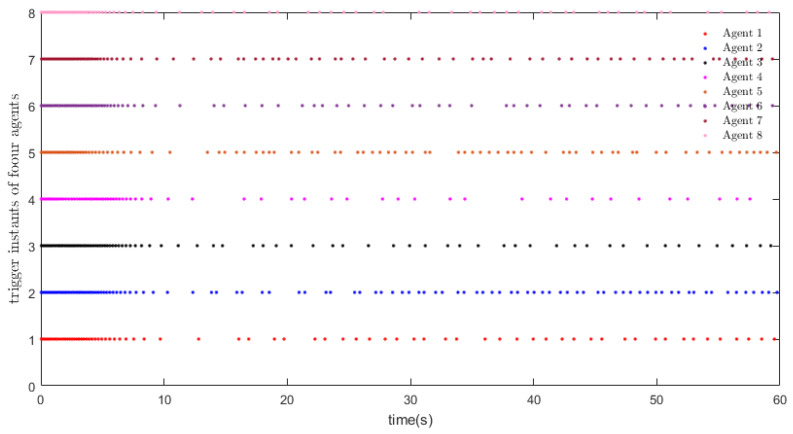
Triggering instants under our strategy.

**Figure 7 entropy-26-00113-f007:**
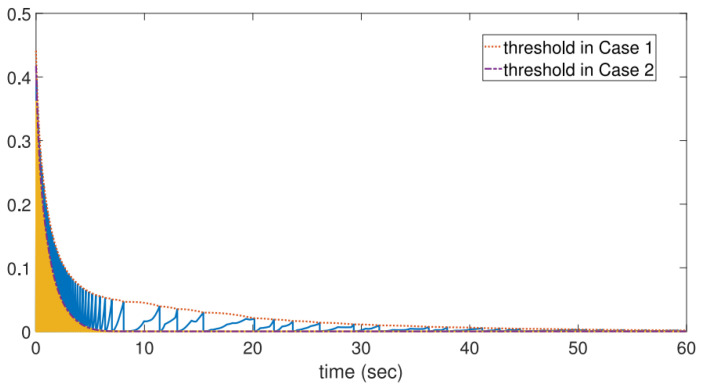
Comparison of triggering threshold and the error’s evolution curves.

**Table 1 entropy-26-00113-t001:** Comparison with static event-triggered approaches.

Control Strategy	$\pi_{i}$	Triggering Numbers for Agents							
		1	2	3	4	5	6	7	8
case1	0.002	78	126	95	65	111	94	105	96
case2	0	584	757	738	630	1062	685	893	786

## Data Availability

Data is contained within the article.

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
