# Peer review of "Distributed Dynamic Event-Triggered Control to Leader-Following Consensus of Nonlinear Multi-Agent Systems with Directed Graphs"

_entropy, 2024, doi:10.3390/e26020113_

Round 1

Reviewer 1 Report (Previous Reviewer 2)

Comments and Suggestions for Authors

In my opinion, the paper can be accepted in its current form.

Author Response

Thank you very much for your review

Reviewer 2 Report (Previous Reviewer 3)

Comments and Suggestions for Authors

I have no more comments, it can be accepted now.

Author Response

Thank you very much for your review

Reviewer 3 Report (New Reviewer)

Comments and Suggestions for Authors

This manuscript deals with the introduction of event-triggered strategies for the design of distributed control laws aimed at multi-agent systems subjected to nonlinear dynamics.

First, from the preparation perspective, the manuscript is poorly written and prepared: the authors should engage a professional editor to address the issues with English. Beyond this particular point, there are numerous other issues, such as punctuation marks in red throughout the manuscript, blank spaces missing, etc.

Second, in their Introduction, the authors are formulating statements that are blatantly wrong. For instance:

* The authors claim that it is commonly assumed that the various agents are interconnected by a network offering enabling continuous signals. This is simply not true. There is a vast and rich literature on the topic of MAS networks with temporal networks, and even static networks with non-continuous signals.

* Regarding event-triggering methods, the authors state that they are meant to replace continous signals by sampled signals. This is not exactly true. Event-triggering approaches are meant to reduce the sampling of signals, from the case of periodic sampling to adaptively modify the sampling according to the needs of the system. The event-triggering law plays that key role.

* On page 2, the authors state that works on event-triggering control for MAS adopt a static event-triggering mechanism. Again that is not true. I encourage the authors to check the more references.

And including works related to cybersecurity in this manuscript.

* Thirdly, the above issues are critical and it is too bad since some of the results in Sec. 3 are interesting. However, the analysis of examples in Sec. 4 is insufficient. The authors are considering trivial problems of leader-follower consensus with only 6 nodes. This too small a system to truly validate the approach.

Comments on the Quality of English Language

First, from the preparation perspective, the manuscript is poorly written and prepared: the authors should engage a professional editor to address the issues with English. Beyond this particular point, there are numerous other issues, such as punctuation marks in red throughout the manuscript, blank spaces missing, etc.

Author Response

Reviewer 4 Report (New Reviewer)

Comments and Suggestions for Authors

This paper proposes a solution for consensus problem, based on a dynamic event-triggered strategy, improving the system by the elimination of Zeno behavior.

The main result is presented in Theorem 1, but equation (12) is not clear. Please revise it because it seems incorrect.

The paper is difficult to read, the presentation must be improved:

a) Revise the written (for instance, in line 39, the sentence begins by a reference; Section 2 and subsection 2.1 have the same title, Assumption 2 is unclear, line 141 ‘we describes’, etc.)

b) Number all the equation, and better present them including spaces to clarify them (line 83, eq (1), etc.)

c) Please better present the Zeno behavior and not say that you include a proof of a definition.

d) Table 1 is not well presented.

Round 2

Reviewer 3 Report (New Reviewer)

Comments and Suggestions for Authors

The authors have not made real efforts to address my concerns.

The literature review is biased and lacks key recent results on exactly this research question.

Comments on the Quality of English Language

The manuscript has to be edited by a native speaker.

Author Response

Reviewer 4 Report (New Reviewer)

Comments and Suggestions for Authors

Accepted.

Author Response

Thank you very much for your thorough review.

This manuscript is a resubmission of an earlier submission. The following is a list of the peer review reports and author responses from that submission.

Round 1

Reviewer 1 Report

Comments and Suggestions for Authors

It appears that the paper has been hastily written, riddled with numerous typos, errors, and incorrect mathematical analyses. Before it can be evaluated for its technical merits, a thorough re-writing is imperative. In its current state, the paper's novelty and contributions hold little significance, while its technical analysis displays grave flaws. A few highlighted concerns include:

  • Equation in line 76 necessitates careful formulation. For instance, if A and D are 3x3 matrices while B and C are 2x2 matrices, compatibility issues arise in the product of A and C.
  • In line 80, did you intend to denote the set of other nodes as {1,2,…, N} instead of {0,1,…,N}?
  • In line 83, is it intended to state “obtained from \tilde{\mathcal{G}}” rather than “obtained from \mathcal{G}”?
  • Why is there a bar on i in the definition of the set \mathcal{N}_i in line 87?
  • Additionally, in line 87, should it not be \omega_{ij} instead of \omega_{ii} for defining \Pi?
  • In line 96, should it be \tilde{G} instead of \hat{G}? If not, clarification on \hat{G} is required.
  • Lemma 1 is incorrect. The product \tilde{\zeta}H is undefined; \tilde{\zeta} is a column vector and H is a matrix, rendering them incompatible for the product.
  • Either prove Lemma 2 or provide a suitable reference.
  • Introduction of q_i and e_i in (4) occurs without prior definition. Either predefine them before usage in (4) or clarify in a sentence like “q_i and e_i are to be defined later in equation (5)”.
  • What is a_ij in (5)?
  • The “i” should be in italics in line 128.
  • The definition of \Theta(x) seems flawed. It should read \Theta(x) = f(x_i) – f(x_0). Alternatively, provide a derivation of (7) in your response letter.
  • In line 133, why does \zeta possess a subscript i?
  • Could you elaborate on (8) in your response letter, detailing its derivation?
  • The origin of (9) needs clarification. How does the solution of (9) correlate with Lemma 1?
  • How does (12) stem from (10)? What makes (11) a suitable contender for a Lyapunov function?
  • The restriction of each x_i as a scalar quantity appears to curtail novelty and significance. I say that x_i is scalar because f_i(x) takes only scalar inputs as mentioned in Assumption 2.
  • (14) is incorrect. The norm on the right-hand side must be inside the summation.
  • Consequently, (15) and ensuing analyses carry doubts. This consequently jeopardizes the validity of theorem 1 and its outcomes.
  • The derivation of the first equality in (21) raises questions. The first expression of || \dot{e}_i || seems flawed.
  • The analysis of Zeno behavior appears hasty and potentially flawed. Particularly, it necessitates validation of why || q_i(t) || adheres to the bound stated in line 182. Without this validation, the entire Zeno behavior analysis loses substance.
Comments on the Quality of English Language

Please verify language usage throughout the manuscript (e.g., line 84).

Line 116 contains two instances of the word “the”.

Reviewer 2 Report

Comments and Suggestions for Authors

please see attached pdf file

Comments on the Quality of English Language

please see attached pdf file (point 15)

Reviewer 3 Report

Comments and Suggestions for Authors

This article studies the leader-following event-triggered consensus of a class of multi-agent systems under nonlinear dynamics. After my reviewing, the work is interesting, but it can not be accepted in its current version and some following comments are required to increase the quality of the article.   

1.  The organization section of this article should be added to make the presentation easy to follow.

2.   The authors claim that a novel dynamic event-triggered mechanism (4) is proposed to reduce the triggering events. However, it has been investigated in many results. What are the main differences of the proposed dynamic ETM from some existing ones? Please give some explanations.

3.  The literature reviewing is insufficient. Some recent results about event-triggered control are required to be discussed to update the reviewing. For example, Sampled memory-event-triggered fuzzy load frequency control for wind power systems subject to outliers and transmission delays, Adaptive memory-event-triggered static output control of T–S fuzzy wind turbine systems.

4.  The figure qualities of some simulation results can be improved.

5.     It is known that time delay is critical issue for communication network. Is it feasible for the proposed event-triggered control strategy to deal with the time delay?

6.     There exist some grammar errors and typos in this paper. For example, in line 236: In further research, we will devote to Probing into the dynamic event-triggered control strategies in countering cyber attacks and time-varying. Please check the presentation carefully.

Comments on the Quality of English Language

Moderate editing of English language required.

Round 2

Reviewer 1 Report

Comments and Suggestions for Authors

I would like to express my gratitude to the authors for their efforts in addressing the comments. However, I must point out that the majority of the concerns raised have not been adequately addressed. I will reiterate these concerns below:

1.      Question 1 has not been addressed adequately. The first equation after (90) does not hold true in general. Restrictions must be imposed on the dimensions of Matrices C and D. For example, consider the case when A and D are 3-by-3 identity matrices, while C and B are 2-by-2 identity matrices. The product in this case is not valid. The authors are expected to exercise necessary caution and ensure mathematical rigor when presenting such statements.

2.      Questions 2-5 have been adequately answered.

3.      Question 6 remains unresolved. It is important to clarify how the product \tilde{\zeta}*H is defined in line 133, especially considering that \tilde{\zeta} is an N-dimensional column vector. Additionally, is \hat{H} a vector or a matrix? What are the dimensions of \hat{H]? It is claimed that \hat{H} is a matrix, but the first term \tilde{\zeta}H is undefined due to a lack of dimensional compatibility, while the second term H^T \tilde{\zeta} results in a vector of dimension N. Therefore, it is incorrect to assert that \hat{H} is a matrix.

4.      Although Question 7 seems to be addressed, could you please provide a specific reference (e.g., theorem/lemma number or section number) to [39] where the proof can be found?

5.      Question 8 has been addressed, but similar typos persist in other places. For example, in (23), it is referred to as a_{ij}, when it should be \varpi_{ij}.

6.      Question 9 has been addressed.

7.      Question 10 remains unanswered. Firstly, the reference (Leader- Following Consensus of Linear Multi-Agent Systems via Dynamic Event-Triggered Adjustable Control Protocol) does not provide any answers as it does not consider the nonlinear term C f(x_i). Secondly, there are several issues with \Theta(x). For instance, why are there transposes of f in line 148 (e.g., the term f^T (x_1) – f^T (x_0))? As previously established, each f is a scalar. Therefore, the meaning of the transpose in this context needs clarification. Moreover, \Theta(x) is of dimension N, while \zeta_i in (7) is a scalar. As a result, the dimensions on the right-hand side of (7) do not match those on the left-hand side, raising doubts about the entire analysis.

8.      Question 11 has been addressed.

9.      Question 12 remains unanswered. The response appears to be a simple copy-paste of the text in the manuscript. Since (7) is incorrect, equation (8) is also incorrect as it is derived from (7).

10.  Question 13 has not been addressed. The source of equation (9) is unclear.

11.  Question 14 has not been addressed. In equation (10), there are two sub-equations. In the first sub-equation, the coefficient of ||q_i||^2 is \delta, while in the second sub-equation, the coefficient is v_i. It is essential to clarify the relationship between \delta and v_i and how (10) ensures the inequality stated in line 163. Furthermore, does the function V in (11) satisfy all the requirements of a Lyapunov function?

12.  The response to Question 15 is unsatisfactory. Could you provide insights obtained from this study that can be applied to the analysis of a vector system where each x_i is a vector? Currently, the significance of the study appears to be only incrementally valuable.

13.  Question 16 has not been addressed. The new equations (14) and (15) appear to be incorrect, with no norm on the right-hand side of (14), which is unexpected. Detailed derivations are required.

14.  It seems that the authors have not identified the issue with (21). Allow me to explain in detail: Based on the definition of e_i(t) in (5), the derivative of e_i(t) is equivalent to the negative derivative of q_i(t), i.e., \dot{e}_i(t) = - \dot{q}_i(t). Since the expression for q_i(t) is provided in the first equation of (5), taking the derivative of both sides is necessary to obtain the derivative of q_i(t). It appears that the authors did not calculate the derivative of q_i(t) and simply substituted the expression of q_i(t) as the expression of \dot{q}_i(t).

15.  While Question 18 has been addressed, the issue remains unresolved. How is the Zeno behavior avoided when \tilde{\zeta}_m = 0? Nowhere in the manuscript is it proven that \tilde{\zeta}_m is strictly positive.

Comments on the Quality of English Language

The language and the presentation can be improved.

Reviewer 2 Report

Comments and Suggestions for Authors I am satisfied with this improved version. I recommend accepting the paper in its current form.

Author Response

Thank you very much for your review

Reviewer 3 Report

Comments and Suggestions for Authors

The comments are replied well, it can be accepted now.

Author Response

Thank you very much for your review

Round 3

Reviewer 1 Report

Comments and Suggestions for Authors

Question 1 is still not addressed correctly. The current manuscript states that “The first equation holds if and only if the number of columns in matrix \tilde{A} and \tilde{C} is equal to the number of rows in matrices \tilde{B} and \tilde{D}”. Consider the case A is 3-by-3, B is 3-by-2, C is 2-by-3 and D is 3-by-2. These matrices satisfy the row column condition mentioned above. However, the product AC is invalid since A and C are not dimensionally compatible for the product.

Question 2 is addressed.

Question 3 is addressed. It might be useful to update the manuscript accordingly, e.g., write [Lemma 6, 39] or [page pp, 39] instead of just [39].

Question 4 seems to be addressed.

A few new concerns appear based on the changes made to the manuscript. For example, in Assumption 2, \alpha_i and \beta_i are denoted as vectors. I am not sure how the authors can have a vector in the denominator when they write (f_i(\alpha_i) – f_i(\beta_i))/(\alpha_i - \beta_i) in line 124. The authors could simply state that f_i is a Lipschitz function and define (3) directly without having to define U_i and L_i.

Moreover, given that \alpha_i is an n dimensional vector, I conclude that x_i is an n dimensional vector since \alpha_i takes the place of x_i in Assumption 2.

The definition q = - (H @ I_N) \zeta (where @ denotes the Kronecker product) does not make sense since the dimensions do not match. Furthermore, \Theta in line 154 does not make sense since each f_i can be a vector and \Theta(x) becomes a matrix.

Question 5 is not addressed, and the current version is equally, if not more, confusing. For example, \zeta_i is not even defined. In the earlier manuscript \zeta_i was defined to be x_i – x_0. The manuscript does not mention anything about the dimensions of B, C, u_i and f_i.

Question 6 is answered. However, the derivation (handwritten) is incorrect. The term (H @ BK) q_i is dimensionally incompatible for the matrix vector multiplication. q_i has n components where as H @ BK has Nn columns.

In line 163, it is stated \delta_i <= \tilde{\zeta}/ \alpha_i. I thought \tilde\zeta is a vector, as defined in Lemma 1. This would make \delta_i a vector.

Question 8: I believe the changes have caused more errors in the manuscript. The analysis needs to be thoroughly checked for dimensional compatibility of matrix multiplications.

I could not follow the response to Question 9. The equations written in that form are not very user friendly to follow. Handwritten calculations are easier to follow than that.

Question 9 (the response letter has two Question 9's) has been partially addressed. I follow the first step now after placing the appropriate derivatives. However, (24) seems to be incorrect. This is because there is no q_j(t^j_k) terms in (24) whereas there were several q_j(t^j_k) terms in (23). Furthermore, \vartheta_ij were multiplied with f_i and there is no \vartheta_ij in (24) and it is simply ||C||L_g. Furthermore, the conversion from f(x_i) – f(x_j) to q_i is not correct.

Question 10 is not answered. Defining \tilde{\zeta}_m > 0 is not enough. As we observed from [39] (the attached screenshot) that \tilde\zeta > 0 for a strongly connected graph. The graph is not strongly connected as there is no path from a node i to the leader 0 (there is only a path from leader 0 to node i).

Comments on the Quality of English Language

The writing can be improved. For example Line 153: "... same cases. and note that ..."